# Exposure to dexamethasone modifies transcriptomic responses of free-living stages of *Strongyloides stercoralis*

**Rutchanee Rodpai**[1], **Oranuch Sanpool**[1], **Tongjit Thanchomnang**[2], **Pokkamol Laoraksawong**[3], **Lakkhana Sadaow**[1], **Patcharaporn Boonroumkaew**[1], **Arporn Wangwiwatsin**[4], **Chaisiri Wongkham**[4], **Porntip Laummaunwai**[1], **Wannaporn Ittiprasert**[5], **Paul J. Brindley**[5], **Pewpan M. Intapan**[1], **Wanchai Maleewong**[1] *

**1** Department of Parasitology, Faculty of Medicine, and Mekong Health Science Research Institute, Khon Kaen University, Khon Kaen, Thailand, **2** Faculty of Medicine, Mahasarakham University, Mahasarakham, Thailand, **3** School of Health Science, Sukhothai Thammathirat Open University, Nonthaburi, Thailand, **4** Department of Biochemistry, Faculty of Medicine, Khon Kaen University, Khon Kaen, Thailand, **5** Department of Microbiology, Immunology and Tropical Medicine, and Research Center for Neglected Diseases of Poverty, School of Medicine and Health Sciences, George Washington University, Washington, DC, United States of America

* wanch_ma@kku.ac.th

**Data Availability Statement:** All sequence reads have been released at the NCBI Sequence Read

## Abstract

Hyperinfection and disseminated infection by the parasitic nematode *Strongyloides stercoralis* can be induced by iatrogenic administration of steroids and immunosuppression and lead to an elevated risk of mortality. Responses of free-living stages of *S. stercoralis* to the therapeutic corticosteroid dexamethasone (DXM) were investigated using RNA-seq transcriptomes of DXM-treated female and male worms. A total of 17,950 genes representing the transcriptome of these free-living adult stages were obtained, among which 199 and 263 were differentially expressed between DXM-treated females and DXM-treated males, respectively, compared with controls. According to Gene Ontology analysis, differentially expressed genes from DXM-treated females participate in developmental process, multicellular organismal process, cell differentiation, carbohydrate metabolic process and embryonic morphogenesis. Others are involved in signaling and signal transduction, including cAMP, cGMP-dependent protein kinase pathway, endocrine system, and thyroid hormone pathway, as based on Kyoto Encyclopedia of Genes and Genomes analysis. The novel findings warrant deeper investigation of the influence of DXM on growth and other pathways in this neglected tropical disease pathogen, particularly in a setting of autoimmune and/or allergic disease, which may require the clinical use of steroid-like hormones during latent or covert strongyloidiasis.

## Introduction

*Strongyloides stercoralis* is a soil-transmitted nematode that infects humans via skin penetration, eventually residing in the small intestine. *S. stercoralis* has a complex developmental cycle

Archive (SRA) under project accession number PRJNA636286.

**Funding:** The grants from Thailand Research Fund, [Distinguished Research Professor grant no. DPG6280002; The Royal Golden Jubilee Ph.D. Program, grant no. PHD/0053/2556] and Khon Kaen University [Research and Graduate Studies Affairs grant no. RP64010; Faculty of Medicine grants no., IN61150 and DR63101]. The funders had no role in study design, data collection and analysis, decision to publish, or preparation of the manuscript.

**Competing interests:** The authors have declared that no competing interests exist.

that exhibits both free-living and parasitic cycles [1]. It is generally difficult to obtain specimens of the parasitic stage, and moreover, an adult male parasitic stage is not known to occur. In contrast, as there are many human cases of strongyloidiasis and larvae can be recovered from feces, access to the free-living cycle of this human pathogen is less difficult. In this situation, the developmental cycle proceeds outside the human host in the external environment, giving rise to free-living forms of both adult male and female worms. First-stage larva excreted in the feces of the host can develop by two possible routes: direct (homogonic) development to an arrested larva (the filariform larva or infective stage larva) and indirect (heterogonic) development to a free-living adult stage. The adults mate and reproduce sexually, leading to release of the egg stage by free-living female worms. These eggs hatch into rhabditiform larvae, which in turn develop into filariform, infective larvae. The filariform larva initiates infection of a human host following direct penetration of skin, leading to the parasitic cycle within the host.

At least 600 million individuals worldwide may be infected with this parasite [2]. Many cases of infection with *S. stercoralis* are mild or asymptomatic. However, in acute infection, a characteristic cutaneous reaction may occur as the larva penetrates the skin, or pulmonary symptoms may be produced as the larva migrates through the lungs [3, 4]. In severe cases, over-proliferation of larvae can lead to hyperinfection and disseminated strongyloidiasis. Dissemination of the larvae to internal organs generally occurs only in high-risk groups, such as coinfection with human T-lymphotropic virus 1 infection (HTLV-1), during malignancy, in the context of alcoholism, and in children. Of particular concern are patients who become immunosuppressed or receive autoimmune disease treatment including corticosteroids [3–5]. With the high prevalence of *S. stercoralis* and the global geographical distribution of this parasite, the risk factors complicate and augment the burden of human strongyloidiasis [6, 7].

Although the molecular mechanisms linking strongyloidiasis with these risk factors remain unclear, the parasite appears to benefit from the suppressed innate and adaptive immune status conferred by concurrent glucocorticoid therapy, resulting in parasite reproduction, skin and tissue invasion and dissemination [3]. Moreover, steroid hormones have been proposed to serve as endogenous ligands of the *S. stercoralis* nuclear hormone receptor DAF-12, which regulates the reproductive process of the nematode [1, 8, 9]. Thus, elevated levels of exogenous corticosteroids may enhance the worm burden, especially during the indirect development route [9].

Dexamethasone (DXM) is a widely used steroid drug for the treatment and control of inflammation [10] and for Covid-19 disease [11]. DXM alters the immune response in animal models, where it is known to inhibit proliferation of eosinophils and mononuclear cells, to induce immunosuppression and to impair innate and adaptive immune responses [10, 12, 13]. DXM also enhances the fertility of the parasitic female *Strongyloides venezuelensis in vivo*, leading to an increased parasite burden, chronic strongyloidiasis, hyperinfection and/or dissemination [10]. Nonetheless, the direct effects of DXM treatment in humans with respect to enhanced fecundity of *S. stercoralis* during both the direct and indirect cycles of development awaits clarification at the transcriptome level.

Recently, expansive databases of transcriptomic data for *S. stercoralis* have become available, including RNA sequences of seven discrete developmental stages of the parasite [14]. In addition, the potential for modulated reproductive expression of *S. stercoralis* genes during steroid treatment has become apparent [1, 8]. Moreover, enhanced fecundity effects on *S. stercoralis* in the environment after exposure to steroids in the human bowel and excretion via feces are worthy of investigation. Such effects may increase the opportunity for population exposure to infective larvae and result in elevated prevalence and incidence in at-risk populations. This study employed a model to mimic iatrogenic exposure of *S. stercoralis* to medicinal steroids in infected humans. The model employed exposure of worms cultured *in vitro* on agar plates to

DXM, followed by investigation of the transcriptome of free-living stages of *S. stercoralis*, including the parental free-living adult male and female stages. Using Illumina-based deep sequencing of total mRNA, we obtained insight into the genomic responses of this pathogenic nematode to this hormone-like agent.

## Methods

### Ethical considerations

The study protocol was approved by the Khon Kaen University Ethics Committee for Human Research (HE611059) with relevant guidelines and regulations of Ethical Principles for Medical Research Involving Human Subjects by the Declaration of Helsinki. Written informed consent was obtained from adult participants and from the parents or legal guardians of minors.

### *Strongyloides stercoralis* worm and agar plate culture

*Strongyloides stercoralis* worms were obtained from the fecal samples of infected asymptomatic patients without corticosteroid treatment. Fecal samples from 30 individual cases of strongyloidiasis were cultured using the agar plate technique [15]. Each stool sample was divided into two groups: one used for the control group (non-DXM) and the other for the test group (DXM). For the DXM group, dexamethasone (approximately 3.6 mg) was spread on a 1.5% nutrient agar plate (9 cm diameter plate) and air dried for 30 min. Thereafter, three grams of stool was placed on the center of the plate; at least three plates/individual fecal samples both with and without the inclusion of DXM were used. Then the plate was maintained for 72 hours at 27˚ C. The free-living adults were harvested individually, washed several times in distilled water, preserved in RNA*later* solution and stored at −20˚ C (S1 Fig). Following this incubation, representative fecal samples of the control group (non-DXM; n = 8, each) and test group (DXM; n = 8, each) were selected for investigation of the numbers of worms according to several developmental stages, i.e., 1) rhabditiform larvae, 2) filariform larvae, 3) free-living adult-stage males, and 4) free-living adult-stage females. The ratios of filariform larvae and the total numbers of free-living adults and filariform larvae were used to calculate the homogonic index [16] and evaluated using the Wilcoxon matched-pairs signed-ranks test in STATA Version 10.1.

### RNA extraction

Total RNA was extracted from each sample (50 μl of worms) of pooled free-living adult males or females that had been preserved in RNA*later* using TRIzol reagent (Invitrogen, Carlsbad, CA) according to the manufacturer's protocol. In brief, the RNA was separated using chloroform, precipitated with isopropanol, and the pellet washed with ethanol and resuspended in RNase-free water. Thereafter, the RNA was incubated with DNase I (New England Biolabs Inc, Ipswich, MA) to remove residual DNA and stored at −70˚ C. The concentration and purity of the RNA was determined at 260 nm (NanoDrop 2000 Spectrophotometer, Thermo Fisher Scientific, Wilmington, DE). The RNA integrity was monitored by agarose gel electrophoresis and staining with ethidium bromide (S2 Fig).

### cDNA synthesis and illumina sequencing

One microgram of total RNA per sample was used to construct each library to be sequenced according to the BGI self-sequencing system (Beijing Genomics Institute (BGI), Guangdong, China). Following extraction of total RNA and incubation in the presence of DNase I, mRNA was isolated using oligo(dT) and fragmented into small pieces. cDNAs were synthesized using

the mRNA fragments as templates. Short cDNA fragments were purified, end-repaired, 3′ adenylated, and ligated to Illumina-compatible adapters. Suitably sized fragments were selected for PCR amplification. During the quality control steps, an Agilent 2100 Bioanalyzer and ABI StepOnePlus Real-Time PCR System were used to quantify and qualify the libraries. Nucleotide sequences of the libraries were determined using the Illumina HiSeq 4000 platform by BGI.

### *De novo* transcriptome assembly-focused methods

Following deep sequencing, low-quality reads of < 15, which constituted > 20% of the read, reads that included > 5% unknown bases (N), and adapters were filtered and removed using the vendor's (BGI) internal software. *De novo* assembly of these transcripts was undertaken using these filtered clean reads and the Trinity program (version: v2.0.6) [17], which combines three independent software modules, Inchworm, Chrysalis, and Butterfly. Thereafter, Tgicl (version: v2.0.6) [18] was used to cluster the transcripts, with the resulting clusters termed unigenes.

To achieve functional annotations, the default parameters of the software tools were used, and unigenes were aligned to the nucleotide sequence database (Nt) (ftp://ftp.ncbi.nlm.nih.gov/blast/db), non-redundant protein sequence database (Nr) (ftp://ftp.ncbi.nlm.nih.gov/blast/db), Cluster of Orthologous Groups of proteins (COG) (http://www.ncbi.nlm.nih.gov/COG), Kyoto Encyclopedia of Genes and Genomes (KEGG) (http://www.genome.jp/kegg), and SwissProt (http://ftp.ebi.ac.uk/pub/databases/swissprot) using BLAST [19]. In addition, further annotations were performed using Blast2GO [20] with Nr annotation to obtain Gene Ontology (GO) (http://geneontology.org) annotations and using InterProScan5 [21] for InterPro annotations (http://www.ebi.ac.uk/interpro).

Following transcriptome assembly, cleaned sequencing data were mapped to unigenes with Bowtie2 [22], and gene expression levels in fragments per kilobase million (FPKM) values were calculated with RSEM [23]. The FPKM heatmap of all unigenes from four individual samples was generated using the pheatmap package in R software [24]. To construct the heatmap, a value of 1 was added to each raw FPKM value prior to $log_2$ transformation to avoid $log_2$ (0). To compare unigene expression levels in two samples, in particular, female control (Fc) versus male control (Mc), female control (Fc) versus female treated (Ft), male control (Mc) versus male treated (Mt), and male treated (Mt) versus female male treated (Ft), unigenes with $log_2$-fold change > 3.00 or < -3.00 and FDR < 0.001 were considered differentially expressed genes (DEGs) by the PoissonDis method [25]. KEGG and GO analyses of DEGs were classified according to official classifications, and functional enrichment was performed using phyper, a function of R software. The *P* value was calculated as a formula in the hypergeometric test and used to calculate the FDR. In general, terms were defined as significantly enriched when FDR was not > 0.001.

### Mapping to the reference genome

The reads were refiltered as follows: low-quality reads with < 20 that constituted > 50% of the read and reads with unknown bases (N) at > 10% of the residues were trimmed using Trimmomatic (version 0.39) [26]. Subsequently, the remaining cleaned reads were mapped to the draft genome sequence of *Strongyloides stercoralis* (genome project PRJEB528, https://parasite.wormbase.org/Strongyloides_stercoralis_prjeb528/Info/Index/) [27] using HISAT2 (version 2.1.0) [28]. Gene expression levels in gene count, transcript count, FPKM, and transcripts per million (TPM) values were calculated with StringTie (version 2.1.1) [29, 30]. From the read count data, differential gene expression analysis was performed using DESeq2 with a blind

protocol. Genes with log$_2$-fold change > 3.00 or < -3.00 were defined as DEGs. In addition, GO enrichment analysis was undertaken using WEGO (http://www.geneontology.org/) [31] and KEGG enrichment pathway analysis using BlastKOALA (https://www.kegg.jp/blastkoala/) [32].

## Results

### Effect of dexamethasone on *Strongyloides stercoralis* during development on agar

Out of 30 individual samples, feces from each of eight strongyloidiasis cases were used for the worm classification in development cycle. At 72 hours of agar plate culture, the worms were classed as free-living males, free-living females, post free-living rhabditiform larvae and post-parasitic filariform larvae of *S. stercoralis*, and the numbers of worms in each class were evaluated using a Wilcoxon matched-pairs signed-ranks test. Median numbers were significantly different between the DXM-treated and non-DXM-treated free-living females and between the rhabditiform larvae exposed or not to DXM ($P < 0.05$ in each case) (Table 1). Moreover, the ratio of rhabditiform larvae to filariform larvae in the DXM treatment group was significantly higher than that in the non-DXM group (Table 1).

### Overview of the transcriptome assembly

Four cDNA libraries were prepared from DXM-treated free-living females (treated females; Ft) or males (treated males; Mt) and untreated free-living females (female control; Fc) or males (male control; Mc) and sequenced using the Illumina paired-end reads approach. In total, 26.42 gigabases were generated for the four libraries following the removal of poor-quality reads. Based on the *de novo* assembly, 65.64, 65.23, 66.74 and 66.71 million clean reads were obtained after filtering of the sequencing raw reads for Fc, Ft, Mc and Mt worms, respectively (Table 2). Overall, we recovered 18,226 unigenes that were subsequently compared against public databases, including NCBI non-redundant protein sequences (Nr), NCBI nucleotide sequences (Nt), protein family annotation (SwissProt), Kyoto Encyclopedia of Genes and Genomes (KEGG), Clusters of Orthologous Groups database (COG), InterPro annotation, and Gene Ontology (GO). Further analysis involved mapping quality clean reads to the

**Table 1. Median numbers of indirect development of human *S. stercoralis* from agar plate cultures maintained for 72 hours in the presence or absence of dexamethasone (DXM).**

|  | Non-DXM treated Median (min:max) | DXM treated Median (min:max) | *P* value | 95%CI |
|---|---|---|---|---|
| Free-living male | 4.34 (1:19.5) | 5.25 (1: 25.25) | 0.23 | -1.5 to 4.22 |
| Free-living female | 11.00 (1:71) | 18.03 (1:100.5) | 0.03* | 1 to 24.38 |
| Free-living female/free-living male ratio | 2.23 (0.4:15.93) | 3.24 (1:7.92) | 0.44 | -3.84 to 1.66 |
| Rhabditiform larva[1] | 832.5 (0: 23415) | 2173.75 (0:32730) | 0.04* | 0.0 to 6654.38 |
| Rhabditiform larva/ free-living female ratio | 48.71 (0:329.79) | 125.42 (0:325.67) | 0.62 | -105.63 to 150.10 |
| Filariform larva[2] | 1985 (0:16215) | 459.69 (0:6090) | 0.07 | -5926.56 to 30 |
| Rhabditiform larva/ filariform larva ratio | 0.67 (0.13:1.44) | 3.58 (0: 5.37) | 0.02* | 2.01 to 4.10 |
| Filariform larva/free-living female ratio | 111.74 (0: 748) | 29.21 (0: 112.5) | 0.05 | -410.79 to 5.65 |
| Homogonic index | 0.99 (0: 1.0) | 0.96 (0: 0.98) | 0.52 | -0.04 to 0.03 |

*$P \leq 0.05$, N = 8

[1] developing *S. stercoralis* post free-living rhabditiform larvae.

[2] developing *S. stercoralis* post parasitic filariform larvae.

**Table 2. Summary statistics for transcriptome analysis by *de novo* assembly and mapping to the *S. stercoralis* genome.**

|  | Female control (Fc) | Female DXM treated (Ft) | Male control (Mc) | Male DXM treated (Mt) |
|---|---|---|---|---|
| Total raw reads (million) | 70.14 | 70.14 | 72.40 | 72.40 |
| *De novo* assembly analysis | | | | |
| Total clean reads (million) | 65.64 | 65.23 | 66.74 | 66.71 |
| Total number of transcripts | 20,443 | 20,775 | 21,139 | 20,239 |
| Mean length of transcripts (nt) | 1066 | 1075 | 1042 | 1029 |
| N50 of transcripts | 1732 | 1744 | 1679 | 1624 |
| GC percentage | 28.34 | 28.34 | 28.19 | 28.29 |
| Total number of unigenes | 15,303 | 15,396 | 16,322 | 15,538 |
| Mean length of unigenes (nt) | 1177 | 1187 | 1137 | 1124 |
| N50 of unigenes | 1781 | 1808 | 1734 | 1672 |
| GC percentage | 28.35 | 28.37 | 28.23 | 28.35 |
| All unigenes = 18,226 (mean length = 1373, N50 = 2084, GC percentage = 28.07) | | | | |
| Reference genome mapping analysis | | | | |
| Total clean reads (million) | 65.60 | 65.20 | 66.69 | 66.67 |
| Total mapped reads (million) | 63.60 | 63.18 | 64.04 | 64.08 |
| GC percentage | 34.02 | 34.12 | 34.34 | 34.53 |
| Number of expressed gene | | | | |
| cutoff > 0 (Total genes = 12,772) | 12,292 | 12,320 | 12,039 | 11,919 |
| cutoff > 3 (Total genes = 10,484) | 8,250 | 8,296 | 7,970 | 7,781 |
| All reference genome mapped genes = 13,098 | | | | |

*S. stercoralis* genome (genome project PRJEB528), with 63.60, 63.18, 64.04 and 64.08 million reads of the Fc, Ft, Mc and Mt worms, respectively, mapped (Table 2). Based on annotation with the reference *S. stercoralis* genome, a total of 10,484 (cutoff > 3) transcribed genes were obtained (Table 2).

## Transcriptome profiles of the free-living adults of *S. stercoralis*

Following *de novo* assembly, clean reads were mapped to unigenes, and the gene expression level for each sample was calculated; 17,950 unigenes were identified (S1 Table), with 16,291, 16,285, 16,288, and 15,892 unigenes being expressed in the Fc, Ft, Mc, and Mt stages, respectively. Many expressed genes were shared among the samples. Specifically, the numbers of expressed genes shared between Fc and Ft, Fc and Mc, Ft and Mt, and Mc and Mt were 15,486, 15,066, 14,775, and 15,220, respectively (Fig 1A). The level of normalized fragments per kilobase million (FPKM) was used to represent transcription levels among the RNA-seq libraries. This revealed gene expression level differences between Ft and Fc and between Mt and Mc (Fig 1B), indicating that exposure to DXM induced transcriptional changes. In addition, notable effects of sex on gene expression patterns were observed, whereby the transcription levels between Fc and Mc worms showed marked differences in gene expression between males and females (Fig 1B).

Similarly, the *S. stercoralis* mapped gene set was used to estimate the gene expression level by counting the reads that mapped to genes. Hierarchical clustering of the sample-to-sample distances from normalized gene counts (using the Rlog function of DESeq2) demonstrated dissimilarities between DXM-induced samples and a large difference between the sexes, as shown by a principal component analysis plot (S3 Fig).

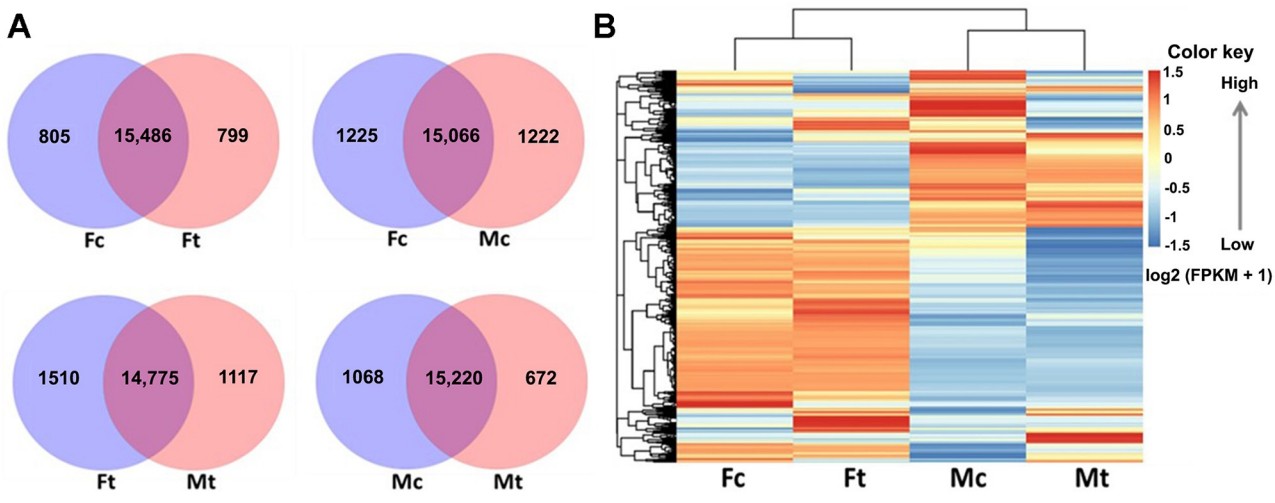

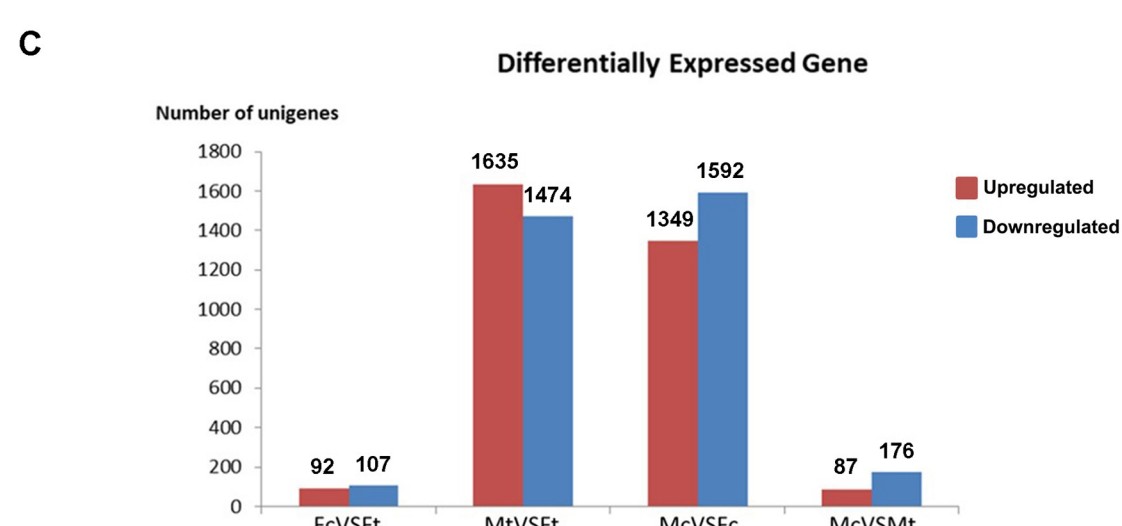

**Fig 1. Differential gene expression pattern of free-living adult *S. stercoralis* based on the *de novo* assembly.** (**A**) Venn diagram showing the number of unique and shared expressed genes among samples. (**B**) Expression profiles of all expressed unigenes are represented in the heatmap, where red and blue indicate higher and lower expression levels, respectively. To construct the heatmap, raw FPKM values were $\log_2$-transformed after a value of 1 was added to the raw value to avoid $\log_2(0)$. (**C**) DEGs were defined as a unigene with $\log_2$-fold change > 3.00 or < -3.00 and FDR < 0.001. DEGs with higher expression levels in Ft than in Fc (FcVSFt), Ft than in Mt (MtVSFt), Fc than in Mc (McVSFc), or Mt than in Mc (McVSMt) were defined as "Upregulated"; those with lower expression levels were defined as "Downregulated" (Fc = female control, Ft = female DXM treated, Mc = male control, Mt = male DXM treated).

### Differential gene expression pattern of free-living male and female adults in response to dexamethasone

Based on the *de novo* assembly, putative differentially expressed genes (DEGs) were identified using the PossionDis algorithms with a cutoff at false discovery rate (FDR) < 0.001 and a $\log_2$ fold change > 3.00 or < -3.00 to identify up- or downregulated DEGs (S2 Table). Employing independence sample analysis and a strict cutoff, four pairwise comparisons were performed to identify DEGs of Fc versus Ft (199 genes identified as DEGs), Mt versus Ft (3,109 genes), Mc versus Fc (2,941 genes), and Mc versus Mt (263 genes), as depicted in Fig 1C.

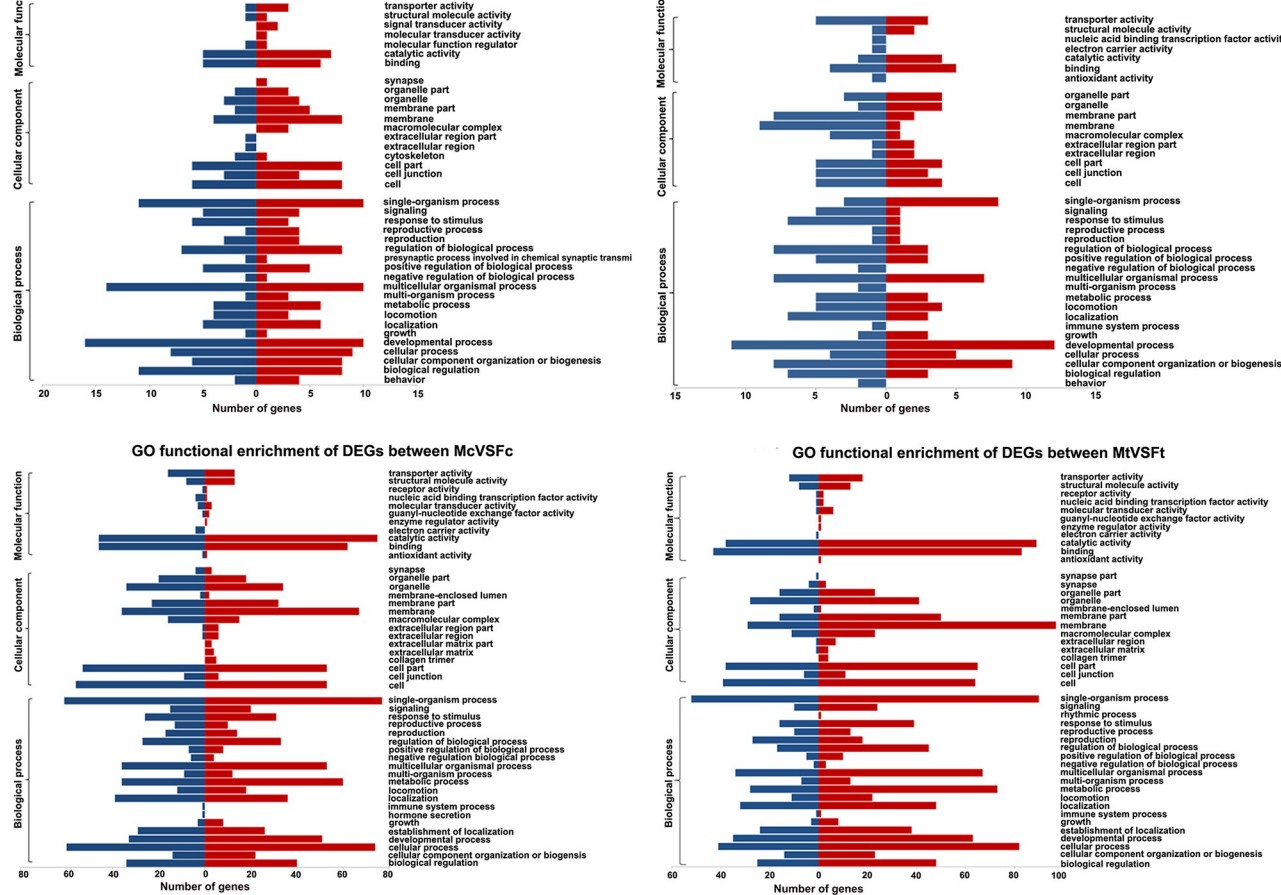

**Fig 2. Gene Ontology (GO) enrichment analysis of differentially expressed genes between samples.** Upregulated genes in enriched GO terms are indicated by red bars, and those with downregulated expression are indicated in blue. FcVSFt; female control versus female DXM treated, MtVSFt; male DXM treated versus female DXM treated, McVSFc; male control versus female control, McVSMt; male control versus male DXM treated.

Based on the reference genome mapping, a cutoff value of $\log_2$ fold change $> 3.00$ or $<$ -3.00 identified DEGs. Comparisons were performed to identify DEGs of Fc versus Ft (536 genes identified as DEGs) and Mc versus Mt (609 genes). Moreover, cluster analysis was applied to identify genes with similar expression patterns under different experimental conditions (Fc, Ft, Mc, and Mt). The top 30 DEGs were obtained (S3 Table). With respect to the sex comparison, a total of 1,943 transcripts were differentially expressed between male and female worms (adjusted *P* value $< 0.05$ and $\log_2$ fold change $> 3.00$ or $< $-3.00) (S4 Table).

## GO classification analysis

To infer the effects of the transcriptional changes, DEGs from the four pairwise comparisons of *de novo* analysis were used for an analysis of GO term enrichment (Fig 2). For Fc versus Ft, enriched GO terms in the biological process category were: *developmental process*, *multicellular organismal process*, *single organism process*, *biological regulation*, *cellular process*, *reproduction*, and *metabolic process*. For Mc versus Mt, the most enriched biological process GO terms were: *cellular component organization or biogenesis*, *developmental process*, and *multicellular-single organismal process*. In contrast, GO terms involved in the regulation of biological process and behavior or response to stimulus seemingly were not enriched among upregulated genes

in DXM-treated males when compared to Fc versus Ft. Accordingly, GO enrichment analysis of DEGs in the mapped reference genome data showed that the most enriched GO terms in the biological process category were similar and included: *biological regulation*, *metabolic process*, *cellular process*, *regulation of biological process*, *response to stimulus*, *signaling*, *cellular component organization or biogenesis*, *positive regulation of biological process*, and *multicellular organismal process* (S4 Fig).

For the sex comparison, male vs female, frequent GO terms observed were: *cellular process*, *developmental process*, *multicellular-single organismal process*, *regulation of biological process*, *biological regulation*, and *metabolic process* (Fig 2; S4 Fig).

## Classification analysis of KEGG pathways

To identify biological pathways in response to DXM treatment, the *de novo* based-DEGs were mapped to the KEGG protein database. Fig 3 presents the pathways in KEGG classified into six main biochemical pathways: organismal systems, metabolism, human diseases, genetic information processing, environmental information processing, and cellular processes. The signal transduction pathway was the most prominent pathway both in comparisons of Fc versus Ft and Mc versus Mt, suggesting that DXM may affect the cell communication of the free-living stages of the worm. Among the KEGG enrichment of DEGs from each comparison, selected significantly enriched pathways are listed in Table 3, with the complete results provided in S5 Table. For Fc versus Ft, free-living females treated with DXM showed DEGs related

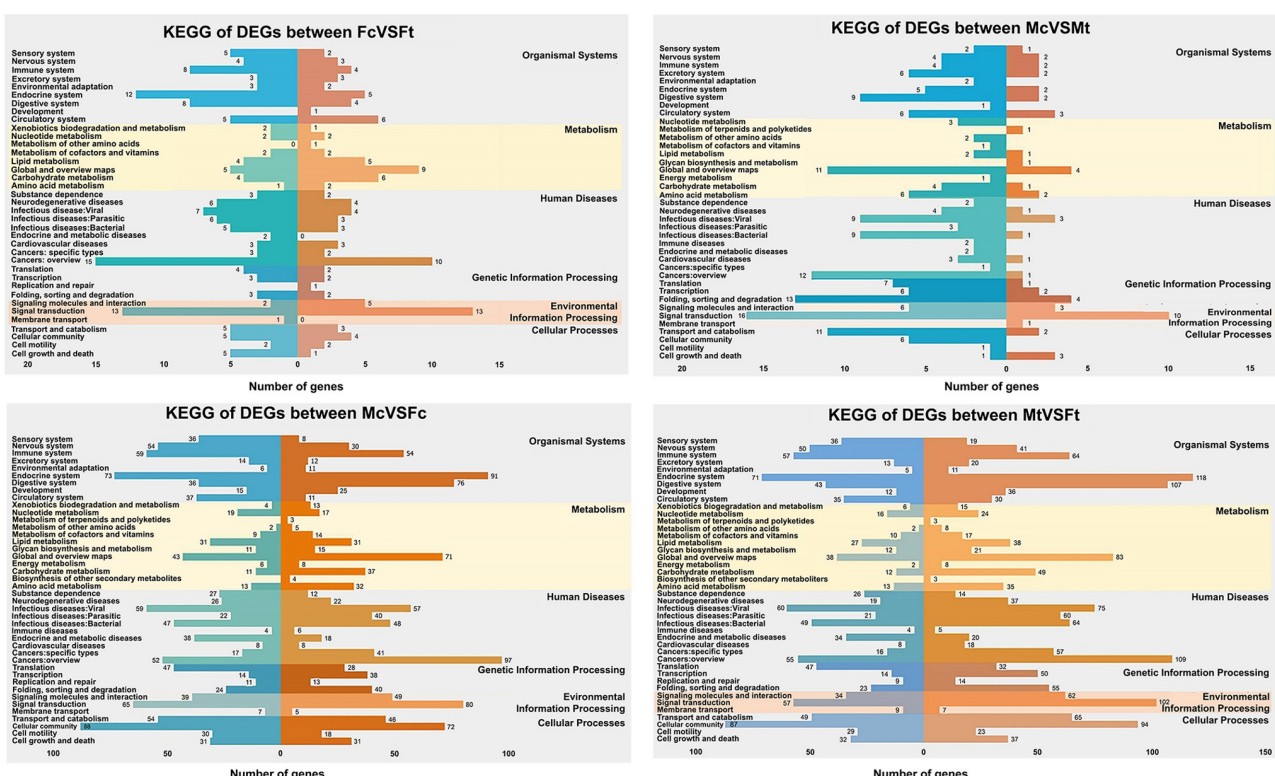

**Fig 3. Kyoto Encyclopedia of Genes and Genomes (KEGG) enrichment analysis of differentially expressed genes (DEGs) between samples.** DEGs that were upregulated are indicated in orange and color gradations of orange, whereas downregulated transcripts are shown in blue and color gradations of blue. FcVSFt; female control versus female DXM treated, MtVSFt; male DXM treated versus female DXM treated, McVSFc; male control versus female control, McVSMt; male control versus male DXM treated.

**Table 3. Selected KEGG pathway analysis from DEGs based on *de novo* assembly.**

| Pathway* | DEGs genes with pathway annotation | All genes with pathway annotation | Pathway ID |
|---|---|---|---|
| FcVSFt | (DEGs = 199) | | |
| cAMP signaling | 7(3.52%) | 372 (3.5%) | ko04024 |
| cGMP-PKG signaling | 6 (3.02%) | 297 (2.79%) | ko04022 |
| Oxytocin signaling | 8(4.02%) | 275 (2.59%) | ko04921 |
| Calcium signaling | 7(3.52%) | 251 (2.36%) | ko04020 |
| Thyroid hormone signaling | 10 (5.03%) | 242 (2.28%) | ko04919 |
| Renin secretion | 6 (3.02%) | 147 (1.38%) | ko04924 |
| Long-term potentiation | 7(3.52%) | 133 (1.25%) | ko04720 |
| Insulin secretion | 5 (2.51%) | 134 (1.26%) | ko04911 |
| Thyroid hormone synthesis | 5 (2.51%) | 114 (1.07%) | ko04918 |
| Endocrine and other factor-regulated calcium reabsorption | 6 (3.02%) | 109 (1.03%) | ko04961 |
| McVSMt | (DEGs = 263) | | |
| ECM-receptor interaction | 4 (1.5%) | 181 (1.7%) | ko04512 |
| Ubiquitin mediated proteolysis | 8 (3.04%) | 153 (1.44%) | ko04120 |
| McVSFc | (DEGs = 2941) | | |
| Pyrimidine metabolism | 20 (0.68%) | 136 (1.28%) | ko00240 |
| Adherens junction | 45 (1.53%) | 160 (1.51%) | ko04520 |
| p53 signaling | 12 (0.41%) | 47 (0.44%) | ko04115 |
| MtVSFt | (DEGs = 3109) | | |
| Spliceosome | 33 (1.06%) | 228 (2.15%) | ko03040 |
| mRNA surveillance | 52 (1.67%) | 168 (1.58%) | ko03015 |
| DNA replication | 9 (0.29%) | 43 (0.4%) | ko03030 |
| Notch signaling | 21 (0.68%) | 85 (0.8%) | ko04330 |
| Adherens junction | 48 (1.54%) | 160 (1.51%) | ko04520 |
| Cell cycle | 33 (1.06%) | 143 (1.35%) | ko04110 |
| Insulin resistance | 44 (1.42%) | 136 (1.28%) | ko04931 |
| Pyrimidine metabolism | 20 (0.64%) | 136 (1.28%) | ko00240 |
| Focal adhesion | 82 (2.64%) | 338 (3.18%) | ko04510 |
| Thyroid hormone signaling | 55 (1.77%) | 242 (2.28%) | ko04919 |

*The pathways have *q*-values $\leq 0.05$ based on genes that are involved in community or signal transformation or the endocrine system and high gene numbers in each pathway.

to signal transduction and the endocrine system. For Mc versus Mt, free-living males treated with DXM exhibited DEGs related to the extracellular matrix (ECM)-receptor interaction pathway and ubiquitin-mediated proteolysis pathway.

Regardless, the most enriched KEGG pathways of the genome mapping based-DEGs were associated with metabolism, including glycosaminoglycan degradation, thiamine metabolism and purine metabolism pathways in free-living females treated with DXM. For free-living males treated with DXM, enriched KEGG pathways included glycosaminoglycan degradation, thiamine metabolism and purine metabolism, drug metabolism-other enzyme pathways, and metabolism of xenobiotics by cytochrome P450. Notably, the latter pathway participates in responses to the external environment, with responses affecting several signal transduction pathways (S5 Fig).

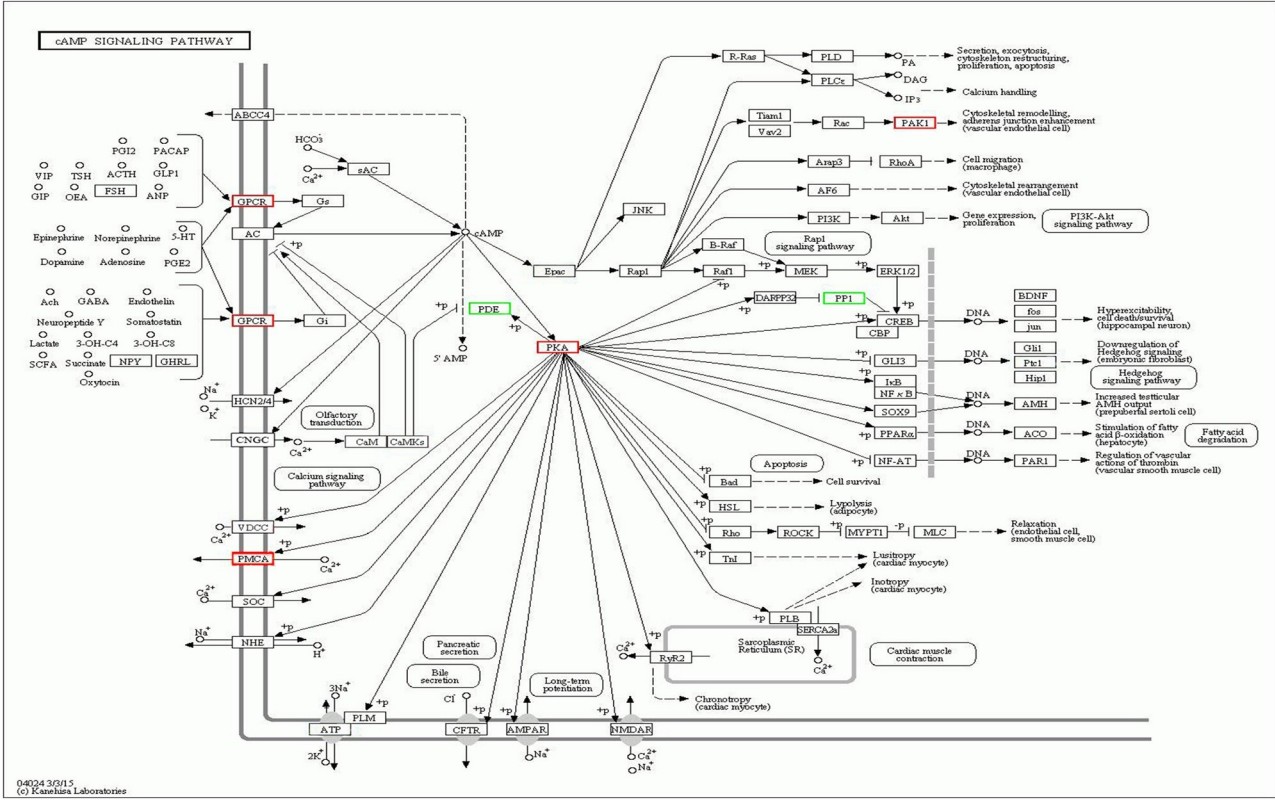

**Fig 4. Up- and down-regulated DEGs in the adenosine 3',5'-cyclic monophosphate (cAMP) signaling pathway (map04024) between female control versus female DXM treated (FcVSFt).** Red boxes indicate upregulated genes in DXM-treated female worms. Green boxes indicate downregulated genes in DXM-treated female worms. (KEGG database network diagram; KEGG website; https://www.genome.jp/dbget-bin/www_bget?map04024).

## Signaling pathways in the free-living adult stage of *S. stercoralis* in response to dexamethasone

KEGG pathway enrichment analysis of DEGs was performed from each comparison. Selected analysis of the most significantly enriched pathways showed that the signaling system was the most significantly enriched pathway ($q$-value $\leq 0.05$) (Table 3; S5 Table). For Fc versus Ft, signaling pathways including cAMP signaling, cGMP-PKG signaling, oxytocin signaling, and calcium signaling were among the enriched pathways. Many genes in the pathways are associated with several unigenes and shared among multiple signaling pathways, as listed in Table 3. For example, protein kinase A (PKA) is a key protein involved in several pathways and is central to the cAMP signaling pathways, as illustrated in Fig 4.

**Accession numbers.** All sequence reads have been deposited in the NCBI Sequence Read Archive (SRA) under project accession number PRJNA636286.

## Discussion

With advancements in sequencing technologies, several *Strongyloides* species have been sequenced and characterized at the transcriptomic level, including *S. stercoralis* [14], *S. venezuelensis* [33], and *S. ratti* [34]. *Strongyloides stercoralis* is an agent of human strongyloidiasis, and its infection can lead to hyperinfection and disseminated strongyloidiasis, particularly in those who are immunosuppressed or undergoing corticosteroid-based treatment for an

autoimmune disease [2, 3]. This study explored the effects of the corticosteroid drug dexamethasone (DXM) on the development of *S. stercoralis* recovered from human infection. We examined the numbers of parasites during four discrete developmental stages in culture on agar and investigated the transcriptomes of the free-living male and female stages of *S. stercoralis*. Exposure to DXM enhanced the numbers of free-living females and rhabditiform larvae in the indirect development phase of the life cycle. The ratio of rhabditiform larvae to filariform larvae in the DXM-treated worms was significantly higher than that in controls. Similarly, it has recently been reported that synthetic dafachronic acid, a class of corticosteroid hormones found in nematodes, also promotes indirect development of *S. stercoralis* [1]. Nevertheless, exposure to DXM was not associated with a significant effect with respect to development stage switching between the direct (homogonic) and indirect (heterogonic) routes of the free-living life cycle of *S. stercoralis* (*P*-value of homogonic index > 0.05). However, the number of free-living males did not significantly change between the DXM-treated group and controls, and the proportion of mean females per males in each group was ~ 3 times greater, a result that prompts further, in-depth genetic studies.

RNA-seq transcriptomic analysis of the parasite was performed to investigate molecular mechanisms that may be responsible for the observed effect on indirect development. To comprehensively compare free-living *S. stercoralis* after DXM treatment with control, non-DXM-treated worms, transcriptomic profiles were examined based on gene expression patterns, key functional annotations, and enrichment analyses of biological processes and pathways. A total of 18,226 unigenes were assembled and then used to calculate gene expression levels, which revealed expression of 17,950 unigenes. Searches were then performed using these unigenes, and annotations in public databases including NT, NR, SwissProt, InterPro, GO, COG, and KEGG databases were retrieved. Several important functions and key pathways associated with the DXM response were then obtained by GO and KEGG enrichment analyses of free-living adult DEGs. Using a stringent $\log_2$-fold change cutoff ($\log_2$FC > 3.00 or < -3.00), we observed varying numbers of up- and downregulated genes across all pairwise comparisons (Fig 1C). These new transcriptome data suggest that the DEGs of free-living females are involved in various signaling pathways, most of which are related to the regulation of development, reproduction, signal hormone transduction and cell division. Not surprisingly, the results suggest markedly different patterns of gene expression between non-DXM- and DXM-treated worms.

In addition, we examined the RNA-seq findings for the *S. stercoralis* genome. A total of 13,098 genes were mapped, and 10,484 (cutoff > 3) transcribed genes were obtained, showing the disadvantage of reference-guided assembly strategies: for instance, more diverged regions may not be covered and may be missing, leading to a lower number present in the target assembly [35, 36]. Nonetheless, the gene expression level pattern was similar to that of the *de novo* assembly analysis, and the top 30 up- and downregulated DEGs of both DXM-treated male and female worms were investigated according to these reference-based genes. Orthologs of genes of the model, free-living nematode *Caenorhabditis elegans* were identified (S3 Table), and among these, *gnrr-7* (human gonadotropin-releasing hormone receptor (GnRHR) related) was upregulated in DXM-treated females. The activity of this receptor is mediated by association with G-proteins that respond to the environment and regulate the reproductive system [37]. In addition, orthologs of genes involved in larval diapause were down-regulated, including *daf-31*, several serpentine receptors, class T gene, and *math-33* [38–40]. For DXM-treated males, upregulated *C. elegans* ortholog genes are related to cell recognition, mechanical stimuli response and signaling process [41–44]. Intriguingly, these included *cyp-29A2* (cytochrome P450 family) orthologs and an ortholog of human *cyp-4A11* (cytochrome P450 family 4 subfamily A member 11), which displays leukotriene-b4 20-monooxygenase activity that

contributes to DXM metabolism in humans [43, 45]. This finding suggests that DXM may induce expression of this *cyp-4A11* ortholog in free-living forms of *S. stercoralis*. Previously, inhibition of cytochrome P450 in *S. stercoralis* was found to affect larval development, with a change in transcript abundance across the developmental cycle [1].

Additionally, the gene expression patterns between females and males exhibited marked differences, but GO and pathway enrichment analyses of DEGs showed a likely symmetric pattern of upregulated unigenes (Fig 2). The most impacted pathways control pyrimidine metabolism, which affects the lifespan of *C. elegans* by regulating reproductive signals [46], and cellular processes, i.e., adherens junctions and the p53 signaling pathway that is involved in embryo morphogenesis and the germ line in *C. elegans* [47, 48] (McVSFc; Table 3). The data suggested that DEGs between females and males are mostly related to development and reproductive regulation. These results can serve as the foundation for future studies on free-living *S. stercoralis* females and males obtained from cultures of infected human feces. In turn, the findings will provide an enhanced understanding of the biology of this helminth pathogen.

Focusing on free-living females, Table 3 shows the 10 most significantly changed pathways. These DEGs are involved in signal transduction and endocrine system pathways that respond to the glucocorticoid DXM, including cAMP signaling, calcium signaling, oxytocin signaling, thyroid hormone signaling, and long-term potentiation. Among these pathways, a shared gene encodes protein kinase A (PKA). Upregulation of PKA, which is a main enzymatic component of the cAMP signaling pathway, is known to influence gene expression, apoptosis, tissue differentiation, and cellular proliferation [49]. As previously reported, DXM can be used as a direct agent to activate PKA and elevate cAMP production to examine downstream regulation mechanisms of the cAMP signaling pathway [50–52]. However, DXM downregulates cyclic-nucleotide phosphodiesterase (PDE) transcription, which has been characterized in rat adipocytes to investigate glucocorticoid-inducing lipolysis via the cAMP-PKA pathway [52]. It is notable that DXM affected free-living *S. stercoralis* females in a manner reminiscent of its impact in vertebrates. Indeed, PKA signaling and function have been characterized in many eukaryotes, including several parasites [53–56]. For example, the cAMP–PKA pathway in *Plasmodium falciparum* has been implicated in gametocyte differentiation [57] and in completion of the asexual cycle as well as reinvasion of erythrocytes [58, 59]. The cAMP-PKA signaling pathway has been implicated in protecting *C. elegans* worms against environmental stress, i.e., low temperatures [60]. In response to DXM, free-living *S. stercoralis* female displayed differential cAMP-PKA signaling pathway gene expression, which resulted in concurrent developmental and reproductive changes. This may enable tolerance to environmental stresses. Moreover, coordination with other signaling pathways was apparent (Table 3, Fig 3); for example, oxytocin signaling and thyroid hormone signaling may enhance growth and development. These signaling pathways play a role in reproductive-related behaviors and promote cell growth, development and differentiation and metabolic processes [61, 62].

For the free-living male, the effects of DXM on development processes were less marked than in the female worm. In fact, the findings for pathway enrichment of DEGs only identified two pathways: i) the ECM-receptor interaction pathway (involved in signaling molecules and interactions) and ii) the ubiquitin-mediated proteolysis pathway (involved in protein folding, sorting and degradation). Genes in these pathways include some significantly upregulated members, though most DEGs were downregulated in both pathways. These findings are intriguing and suggest that further studies on the influence of corticosteroid-like hormones on the EMC-receptor interaction and/ubiquitin-mediated proteolysis pathway in free-living *S. stercoralis* males will be informative. This report, nevertheless, provides the first transcriptomic profile of the free-living male stage of *S. stercoralis*.

The findings from this study confirmed that the glucocorticoid dexamethasone modifies developmental processes in the free-living stages of *S. stercoralis*. The changes observed in response to DXM in developmental pathways and the genes involved enhance our understanding of the biology of free-living *S. stercoralis* and the relationship between steroid-based therapy and increasing people at-risk by this pathogen. It will be valuable to investigate comparative effects by DXM and other corticosteroids on specific genes, specifically with regard to sex regulation, developmental pathways of free-living *S. stercoralis* stages, and DEGs in free-living females involved in various signaling pathways, the regulation of development-related processes, reproduction, signal hormone transduction and cell division. The lacking data and biological replicates need to be clarified in the further investigations. These types of studies should enhance our knowledge of the effects of steroid hormones on parasite biology, including on the heterogonic cycle of development, and enhance our understanding of the global spread of human strongyloidiasis and of the influence of environmental factors, including temperature, humidity, personal hygiene, and steroid drug use.

Further investigation on the effects of DXM on this pathogen with respect to iatrogenic administration of glucocorticoids during treatment of autoimmune diseases or allergies in patients at risk of latent infection with *S. stercoralis* is warranted. In general, newly described functional genomics approaches for investigation of the biology and pathophysiology of helminthiasis will be informative [63–66].

## Supporting information

**S1 Fig. A schematic diagram of the cultures and treatment conditions.**
(TIF)

**S2 Fig. Agarose gel electrophoresis pattern of total RNA extracted.** Left, the total RNA pattern extracted from pooled free-living adult male control (Control), free-living adult male treated with DXM (Test). Right, the total RNA pattern extracted free-living adult female control (Control)and free-living adult female treated with DXM (Test). MW, DNA ladder marker.
(TIF)

**S3 Fig. Principal component analysis (PCA) on the read counts of free-living adult *S. stercoralis* with or without dexamethasone treated base on the reference genome mapping analysis.**
(TIF)

**S4 Fig. Gene ontology (GO) of Fc versus Ft and Mc versus Mt targeted DEGs by WEGO 2.0.** (a) The WEGO histogram of Fc versus Ft and Mc versus Mt targeted DEGs. The x-axis displays the GO terms. The right y-axis shows the gene numbers, while the left y-axis shows the percentages. (b) Log of P-values of GO terms indicated the significant differences between the down and up-regulated genes.
(TIF)

**S5 Fig. Metabolism of xenobiotics by cytochrome P450 pathway (map00980) of the free-living male.** Red boxes indicate significant genes in DXM-treated male worms.
(TIF)

**S1 Table. All gene expression from *de novo* assembly analysis.**
(XLSX)

**S2 Table. Differentially expressed genes between sample based on the *de novo* genome assembly.** FcVSFt; female control versus female DXM treated, MtVSFt; male DXM treated

versus female DXM treated, McVSFc; male control versus female control, McVSMt; male control versus male DXM treated.
(XLSX)

**S3 Table. The top 30 differentially expressed genes by cluster analysis base on the reference genome mapping analysis.**
(PDF)

**S4 Table. Differentially expressed genes between male and female based on the reference genome mapping.**
(XLSX)

**S5 Table. List of significantly enriched KEGG pathways of DEGs.**
(XLSX)

## Author Contributions

**Conceptualization:** Rutchanee Rodpai, Oranuch Sanpool, Tongjit Thanchomnang, Pokkamol Laoraksawong, Lakkhana Sadaow, Patcharaporn Boonroumkaew, Arporn Wangwiwatsin, Chaisiri Wongkham, Porntip Laummaunwai, Wannaporn Ittiprasert, Paul J. Brindley, Pewpan M. Intapan, Wanchai Maleewong.

**Data curation:** Rutchanee Rodpai, Oranuch Sanpool, Tongjit Thanchomnang, Pokkamol Laoraksawong, Lakkhana Sadaow, Patcharaporn Boonroumkaew, Arporn Wangwiwatsin, Chaisiri Wongkham, Porntip Laummaunwai, Wannaporn Ittiprasert, Paul J. Brindley, Pewpan M. Intapan, Wanchai Maleewong.

**Formal analysis:** Rutchanee Rodpai, Oranuch Sanpool, Tongjit Thanchomnang, Pokkamol Laoraksawong, Lakkhana Sadaow, Patcharaporn Boonroumkaew, Arporn Wangwiwatsin, Chaisiri Wongkham, Porntip Laummaunwai, Wannaporn Ittiprasert, Paul J. Brindley, Pewpan M. Intapan, Wanchai Maleewong.

**Funding acquisition:** Pewpan M. Intapan, Wanchai Maleewong.

**Methodology:** Rutchanee Rodpai, Oranuch Sanpool, Tongjit Thanchomnang, Pokkamol Laoraksawong, Lakkhana Sadaow, Patcharaporn Boonroumkaew.

**Project administration:** Wanchai Maleewong.

**Supervision:** Chaisiri Wongkham, Paul J. Brindley, Pewpan M. Intapan, Wanchai Maleewong.

**Writing – original draft:** Rutchanee Rodpai, Paul J. Brindley, Wanchai Maleewong.

**Writing – review & editing:** Rutchanee Rodpai, Oranuch Sanpool, Tongjit Thanchomnang, Pokkamol Laoraksawong, Lakkhana Sadaow, Patcharaporn Boonroumkaew, Arporn Wangwiwatsin, Chaisiri Wongkham, Porntip Laummaunwai, Wannaporn Ittiprasert, Paul J. Brindley, Pewpan M. Intapan, Wanchai Maleewong.

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
