## [Decision Letter · Decision Letter 0]

15 Apr 2021

PONE-D-21-00163

Exposure to dexamethasone modifies transcriptomic responses of free-living stages of Strongyloides stercoralis

PLOS ONE

Dear Dr. Maleewong,

Thank you for submitting your manuscript to PLOS ONE. After careful consideration, we feel that it has merit but does not fully meet PLOS ONE’s publication criteria as it currently stands. Therefore, we invite you to submit a revised version of the manuscript that addresses the points raised during the review process.

The reviewers have brought up several serious concerns. The foremost being a concern about developmental timing in addition to a major concern regarding the use of environmental stages and their relevance to the parasitic stages and steroid-based therapy. Please carefully consider the reviewers comments and address them point-by-point.

We look forward to receiving your revised manuscript.

Kind regards,

Adler R. Dillman, Ph.D.

Academic Editor

PLOS ONE

Journal Requirements:

Please provide additional details regarding participant consent. In the ethics statement in the Methods and online submission information, please ensure that you have specified what type you obtained (for instance, written or verbal, and if verbal, how it was documented and witnessed). If your study included minors, state whether you obtained consent from parents or guardians. If the need for consent was waived by the ethics committee, please include this information.

PLOS ONE now requires that authors provide the original uncropped and unadjusted images underlying all blot or gel results reported in a submission’s figures or Supporting Information files. This policy and the journal’s other requirements for blot/gel reporting and figure preparation are described in detail at https://journals.plos.org/plosone/s/figures#loc-blot-and-gel-reporting-requirements and https://journals.plos.org/plosone/s/figures#loc-preparing-figures-from-image-files. When you submit your revised manuscript, please ensure that your figures adhere fully to these guidelines and provide the original underlying images for all blot or gel data reported in your submission. See the following link for instructions on providing the original image data: https://journals.plos.org/plosone/s/figures#loc-original-images-for-blots-and-gels.

Thank you for stating the following financial disclosure:

4a)         Please clarify the sources of funding (financial or material support) for your study. List the grants or organizations that supported your study, including funding received from your institution.

4b)         State what role the funders took in the study. If the funders had no role in your study, please state: “The funders had no role in study design, data collection and analysis, decision to publish, or preparation of the manuscript.”

4c)          If any authors received a salary from any of your funders, please state which authors and which funders.

4d)         If you did not receive any funding for this study, please state: “The authors received no specific funding for this work.”

We note that you have stated that you will provide repository information for your data at acceptance. Should your manuscript be accepted for publication, we will hold it until you provide the relevant accession numbers or DOIs necessary to access your data. If you wish to make changes to your Data Availability statement, please describe these changes in your cover letter and we will update your Data Availability statement to reflect the information you provide.

Additional Editor Comments:

The reviewers have brought up several serious concerns. The foremost being a concern about developmental timing in addition to a major concern regarding the use of environmental stages and their relevance to the parasitic stages and steroid-based therapy. Please carefully consider the reviewers comments and address them point-by-point.

Reviewers' comments:

Reviewer's Responses to Questions

**Comments to the Author**

1. Is the manuscript technically sound, and do the data support the conclusions?

Reviewer #1: Partly

Reviewer #2: No

2. Has the statistical analysis been performed appropriately and rigorously? 

Reviewer #1: Yes

Reviewer #2: I Don't Know

3. Have the authors made all data underlying the findings in their manuscript fully available?

Reviewer #1: Yes

Reviewer #2: Yes

4. Is the manuscript presented in an intelligible fashion and written in standard English?

Reviewer #1: Yes

Reviewer #2: No

5. Review Comments to the Author

Reviewer #1: The authors of ” Exposure to dexamethasone modifies transcriptomic responses of free-living stages of Strongyloides stercoralis” demonstrate that the free living stages of the S. stercoralis lifecycle do appear to respond to the presence of dexamethasone via the regulation of multiple different genes. While the data does not add significant understanding to the interaction between steroids and the parasitic stages, it does provide evidence that at least this life cycle stage can respond to the steroid.

Major Comments

1) Line 44 – While it is intriguing to discover that free living S. stercoralis do respond to the presence of dex they unlikely to encounter dex in the environment. With both the gerbil and NSG animal models available the recovery of the parasitic stages if possible and would provide a more accurate representation of the response of S. stercoralis to steroids. Line 462-463 states that it helps define the relationship between “steroid-based therapy”. Hyper infection and disseminated strongyloidiasis are due to the presence of corticosteroids and the parasitic stages of S. stercoralis. Please further explain how these results from the free-living stages applies to the understanding of the relationship between steroids and the parasitic stages. What was the rational for doing the analysis on free living stages

2) How was the concentration of Dex determined. Was it a physiological concentration that is found in feces after a patients treatment with the steroids? The author only gives 3.6mg per plate and no total volume so the reader is unable to determine the exact concentration.

3) While the authors did use multiple plates/individual and made multiple cDNA libraries from the parasites collected from the plates at 72 hours it appears that they growth and differentiation of the parasites isolated from the feces on agar plates was not done as a true replicate.

Reviewer #2: In this paper, the authors collect fecal samples from humans infected with Strongyloides stercoralis, incubate the feces on agar plates either in the presence or absence of the corticosteroid dexamethasone (DXM), and then look for transcriptional differences resulting from dexamethasone exposure. Corticosteroid treatment in infected individuals can result in disseminated strongyloidiasis, and while this paper looks at DXM exposure during the free-living rather than parasitic phase of the life cycle, the paper still has important implications for human health. A better understanding of how DXM affects worm growth and physiology could inform nematode control strategies. However, I have a number of issues with the experimental design of this study that in my opinion make the data difficult to interpret.

My major concern is whether DXM is affecting the timing of Strongyloides development. This study looks at a single time point. The authors count the number of male adults, female adults, iL3s, and free-living larvae present in the control vs. DXM-treated samples, but they do not examine the worms by DIC microscopy to determine, for example, whether the adults in the control vs. DXM-treated samples appear to be the same developmental age. A lot of the transcriptional changes the authors observe in control vs. DXM-treated samples have to do with “regulation of development, reproduction, signal hormone transduction, and cell division” (Line 391). These differences could arise due to a direct effect of DXM, or they could arise because DXM exposure either speeds up or slows down development. Without a careful analysis of development in the control vs. DXM-treated samples, and without looking at multiple time points, it’s impossible to distinguish between these possibilities.

Other comments:

1. More explanation of why the 72 h time point was chosen is needed. In lines 196-198, the authors say, “After 72 hours of agar plate culture, the worms were classed as free-living males, free-living females, post free-living rhabditiform larvae and post-parasitic filariform larvae of S. stercoralis.” How do the authors know that the iL3s are post-parasitic? Isn’t 72 h enough time for some of the post-free-living larvae to develop into iL3s? If DXM does alter developmental timing, isn’t this especially a concern for the DXM-treated worms?

2. Lines 193-196: “Fecal samples from 30 individual cases of strongyloidiasis were cultured using the agar plate technique [30] with and without dexamethasone (DXM); at least three plates/individual fecal samples both with and without the inclusion of DXM were used (S2 Fig). Feces from each of eight strongyloidiasis cases were used for the investigation.” This is confusing. Were feces from 30 individuals or 8 individuals used?

3. The first paragraph of the discussion states that DXM significantly affects the number of worms in the indirect phase of the life cycle, and then later states that DXM does not have a significant effect on whether worms enter the direct vs. indirect phase of the life cycle. The wording here is very confusing and needs to be clarified.

4. Line 57: 100 million seems like an outdated estimate of the number of individuals infected with Strongyloides. Haven’t more recent estimates put the number closer to 600 million?

5. The paper would benefit from careful editing. Some of the sentences don’t entirely make sense as written, such as this one: “However, the experiment was done without replicates, did not validate in worm genetic was because of different human host and environment, also confirming the statistical comparisons analysis.” I’m not sure I follow what the authors are trying to say here. Some grammatical edits are required for the figures as well, and also for the short title.

6. PLOS authors have the option to publish the peer review history of their article (what does this mean?). If published, this will include your full peer review and any attached files.

Reviewer #1: No

Reviewer #2: No

---

## [Author Response · Author response to Decision Letter 0]

30 May 2021

Point-by-point response to the reviewers’ comments

PONE-D-21-00163

Exposure to dexamethasone modifies transcriptomic responses of free-living stages of Strongyloides stercoralis

PLOS ONE

Additional Editor Comments:

The reviewers have brought up several serious concerns. The foremost being a concern about developmental timing in addition to a major concern regarding the use of environmental stages and their relevance to the parasitic stages and steroid-based therapy. Please carefully consider the reviewers comments and address them point-by-point.

Reply: We thank you very much. Our responses to reviewers’ comments are provided below in blue ink.

Reviewer #1: The authors of “Exposure to dexamethasone modifies transcriptomic responses of free-living stages of Strongyloides stercoralis” demonstrate that the free living stages of the S. stercoralis lifecycle do appear to respond to the presence of dexamethasone via the regulation of multiple different genes. While the data does not add significant understanding to the interaction between steroids and the parasitic stages, it does provide evidence that at least this life cycle stage can respond to the steroid.

Major Comments

1) Line 44 – While it is intriguing to discover that free living S. stercoralis do respond to the presence of dex they unlikely to encounter dex in the environment. With both the gerbil and NSG animal models available the recovery of the parasitic stages if possible and would provide a more accurate representation of the response of S. stercoralis to steroids. 

Reply: This study aimed to examine fecundity effects on S. stercoralis in the environment after exposure to steroids in the human bowel and excretion via feces. Such enhanced effects may increase the opportunity for population exposure to infective larvae. If our hypothesis is true, regarding to the dexamethasone interferes the human immune system, also the dexamethasone has effect to worms and enhance reproductive growth, increased the worm burden and caused hyper-infection and passed the worms through feces to the environment. Moreover, the dexamethasone may still affect to the free-living worm that enhances worm fecundity and releasing more infective larvae transmitted to humans. This could, in turn, enhance the risk for high prevalence and infection rate into human populations. This study employed a model to mimic iatrogenic exposure of S. stercoralis to medicinal steroids in bowel of infected humans after expose to dexamethasone. To address the hypothesis, worms in feces were exposed to dexamethasone after which we investigated its impact on the transcriptome of the free-living adult stage. For clarity, we modified one sentence, lines 92-93, to “This study employed a model to mimic iatrogenic exposure of S. stercoralis to medicinal steroids in infected humans.” 

Line 462-463 states that it helps define the relationship between “steroid-based therapy”. Hyper infection and disseminated strongyloidiasis are due to the presence of corticosteroids and the parasitic stages of S. stercoralis. Please further explain how these results from the free-living stages applies to the understanding of the relationship between steroids and the parasitic stages. What was the rational for doing the analysis on free living stages

Reply: We meant “The changes observed in response to DXM in developmental pathways and the genes involved enhance our understanding of the biology of free-living S. stercoralis and possibly explain the relationship between steroid-based therapy and increased opportunity for population exposure to infective larvae, which may in turn lead to elevated prevalence and incidence in at-risk populations. 

For clarity, we modified the sentence to “The changes observed in response to DXM in developmental pathways and the genes involved enhance our understanding of the biology of free-living S. stercoralis and the relationship between steroid-based therapy and increasing people at-risk parasitism by this pathogen.” Please see lines 458-461.

2) How was the concentration of Dex determined. Was it a physiological concentration that is found in feces after a patients treatment with the steroids? The author only gives 3.6mg per plate and no total volume so the reader is unable to determine the exact concentration.

Reply: This is the approximate concentration from previous reports in experimental animals. Varying DXM concentration excreted in feces have been reported, within 96 hours, 44% of DXM dose will excrete in rat feces (Rice et al., 1974), whereas English et al. (1975) reported that within 96 hours, 24.8% of DXM dose in rats will have been excretes via the feces. In bovine feces, 95% was eliminated within 3 days (Vanhaecke et al. 2011).

Therefore, in this study, the DXM was covered on the agar. We used 3.6 mg DXM spread on 15 ml-settled agar, an amount calculated based on high-dose DXM (40mg/day) (Gutiérrez-Espíndola et al. 2003) and fecal excretion per day in a health person -- approximately 200 grams (Cummings et al. 1992). Thus, we employed 3.6 mg DXM, which mimics DXM exposure in the human gut lumen, equivalent to ([40 mg DXM/200 g feces] x 18 g (agar (15g)+feces (3g)) = 3.6 mg).

References

1. English J et al (1975). The metabolism of dexamethasone in the rat--effect of phenytoin. J Steroid Biochem. 6(1):65-8. doi: 10.1016/0022-4731(75)90030-8.

2. Rice et al. (1974) The metabolism of dexamethasone in the rat. Biochem. Soc. Transact., 53rd Meeting, Bristol, 2: 107-109.

3. Vanhaecke et al. (2011) Elimination kinetics of dexamethasone in bovine urine, hair and feces following single administration of dexamethasone acetate and phosphate esters. Steroids. 76(1-2):111-7. 

4. Gutiérrez-Espíndola et al. (2003). High Doses of Dexamethasone in Adult Patients with Idiopathic Thrombocytopenic Purpura. Archives of Medical Research, 34(1), 31–34.

5. Cummings et al. (1992). Fecal weight, colon cancer risk, and dietary intake of nonstarch polysaccharides (dietary fiber). Gastroenterology. 103(6):1783-9. doi: 10.1016/0016-5085(92)91435-7.

3) While the authors did use multiple plates/individual and made multiple cDNA libraries from the parasites collected from the plates at 72 hours it appears that they growth and differentiation of the parasites isolated from the feces on agar plates was not done as a true replicate.

Reply: We concur with the reviewer. The worms were collected from S. stercoralis infected persons. We have limited sample collections for RNA-S. stercoralis preparations. However, to clarify this point, we used the pooled worm samples from 30 patients and evaluation using the high cut-off value of gene expression level and analyzed with both the de novo assembly and genome mapping analysis to address this issue. We can aim to undertake the replicates recommended by the reviewer in future studies.

Reviewer #2: In this paper, the authors collect fecal samples from humans infected with Strongyloides stercoralis, incubate the feces on agar plates either in the presence or absence of the corticosteroid dexamethasone (DXM), and then look for transcriptional differences resulting from dexamethasone exposure. Corticosteroid treatment in infected individuals can result in disseminated strongyloidiasis, and while this paper looks at DXM exposure during the free-living rather than parasitic phase of the life cycle, the paper still has important implications for human health. A better understanding of how DXM affects worm growth and physiology could inform nematode control strategies. However, I have a number of issues with the experimental design of this study that in my opinion make the data difficult to interpret.

My major concern is whether DXM is affecting the timing of Strongyloides development. This study looks at a single time point. The authors count the number of male adults, female adults, iL3s, and free-living larvae present in the control vs. DXM-treated samples, but they do not examine the worms by DIC microscopy to determine, for example, whether the adults in the control vs. DXM-treated samples appear to be the same developmental age. A lot of the transcriptional changes the authors observe in control vs. DXM-treated samples have to do with “regulation of development, reproduction, signal hormone transduction, and cell division” (Line 391). These differences could arise due to a direct effect of DXM, or they could arise because DXM exposure either speeds up or slows down development. Without a careful analysis of development in the control vs. DXM-treated samples, and without looking at multiple time points, it’s impossible to distinguish between these possibilities.

Reply: We agree with the reviewer. These are the limitations of our study. However, to clarify these points, we used pooled worm samples from 30 patients and evaluation using the high cut-off value of gene expression level and analyzed with both the de novo assembly and genome mapping analysis to solve this problem. We can aim to undertake a multiple time points experiment in future studies.

Other comments:

1. More explanation of why the 72 h time point was chosen is needed. In lines 196-198, the authors say, “After 72 hours of agar plate culture, the worms were classed as free-living males, free-living females, post free-living rhabditiform larvae and post-parasitic filariform larvae of S. stercoralis.” How do the authors know that the iL3s are post-parasitic? Isn’t 72 h enough time for some of the post-free-living larvae to develop into iL3s? If DXM does alter developmental timing, isn’t this especially a concern for the DXM-treated worms?

Reply: At lower temperatures (20-28°C) than the human body (37°C), the free-living life cycle of Strongyloides develops completely from L1 to iL3 via indirect development for >72 hr (3-5 day) (Lok 2007). The first post-free-living L1 is present for about 48 hr (Lok 2007) whereas by 72 hr, it is presumed that the larvae still be in the post-free-living L2 stage. However, the developmental time points of the free-living life cycle of Strongyloides effected by DXM need further study. However, DXM does not significantly affect the number of filariform larvae (Table 1).

Lok, J.B. Strongyloides stercoralis: a model for translational research on parasitic nematode biology (February 17, 2007), WormBook, ed. The C. elegans Research Community, WormBook, doi/10.1895/wormbook.1.134.1, http://www.wormbook.org.

2. Lines 193-196: “Fecal samples from 30 individual cases of strongyloidiasis were cultured using the agar plate technique [30] with and without dexamethasone (DXM); at least three plates/individual fecal samples both with and without the inclusion of DXM were used (S2 Fig). Feces from each of eight strongyloidiasis cases were used for the investigation.” This is confusing. Were feces from 30 individuals or 8 individuals used?

Reply: Revised as recommended: results, page 9, lines 196 to 197; materials and methods, page 5, lines 107-114.

3. The first paragraph of the discussion states that DXM significantly affects the number of worms in the indirect phase of the life cycle, and then later states that DXM does not have a significant effect on whether worms enter the direct vs. indirect phase of the life cycle. The wording here is very confusing and needs to be clarified.

Reply: The initial sentences discussed how DXM significantly affects the number of free-living females and rhabditiform larvae in the indirect development phase of the life cycle; the ratio of rhabditiform larvae per filariform larvae in the DXM-treated worms was significantly higher than that in controls. The latter sentences dealt with the switching of the route to develop in the free-living life cycle that dexamethasone is not significantly affecting. The life cycle switching represents by homogonic index which is if the value nearly zero, it means that they are switching the route to heterogonic (indirect) development. Therefore, aiming for clarity, we have added “(P-value of homogonic index > 0.05)”. (page 20, lines 372).

4. Line 57: 100 million seems like an outdated estimate of the number of individuals infected with Strongyloides. Haven’t more recent estimates put the number closer to 600 million?

Reply: We have changed to reference [2] Buonfrate et al 2020, please see at page 3, lines 57. 

5. The paper would benefit from careful editing. Some of the sentences don’t entirely make sense as written, such as this one: “However, the experiment was done without replicates, did not validate in worm genetic was because of different human host and environment, also confirming the statistical comparisons analysis.” I’m not sure I follow what the authors are trying to say here. Some grammatical edits are required for the figures as well, and also for the short title.

Reply: The meaning was similar that of the next sentence. For clarity, we deleted “However, the experiment was done without replicates, did not validate in worm genetic was because of different human host and environment, also confirming the statistical comparisons analysis.” 

Finally, we would like to thank the reviewers for your kind suggestions.

---

## [Editor Report · Decision Letter 1]

11 Jun 2021

Exposure to dexamethasone modifies transcriptomic responses of free-living stages of Strongyloides stercoralis

PONE-D-21-00163R1

Dear Dr. Maleewong,

We’re pleased to inform you that your manuscript has been judged scientifically suitable for publication and will be formally accepted for publication once it meets all outstanding technical requirements.

Kind regards,

Adler R. Dillman, Ph.D.

Academic Editor

PLOS ONE

Additional Editor Comments (optional):

Thank you for revising the manuscript to address the reviewer's concerns. This study contributes to our understanding of DXM-strongyloides interactions and should be of interest to the field.
---

## [Editor Report · Acceptance letter]

18 Jun 2021

PONE-D-21-00163R1 

Exposure to dexamethasone modifies transcriptomic responses of free-living stages of *Strongyloides stercoralis*

Dear Dr. Maleewong:

I'm pleased to inform you that your manuscript has been deemed suitable for publication in PLOS ONE. Congratulations! Your manuscript is now with our production department. 

Kind regards, 

on behalf of

Dr. Adler R. Dillman 

Academic Editor

PLOS ONE